# REVISITING ASSOCIATIVE RECALL IN MODERN RECURRENT MODELS

## ABSTRACT

Modern recurrent deep learning models – such as state-space models (SSMs) – have emerged as a promising computationally efficient alternative to Transformers for sequence modeling. However, how their practical differences in learnability and optimization impact core capabilities remains underexplored. In this paper, we thoroughly compare SSM and Transformer learning dynamics on two fundamental benchmarks highly correlated with language modeling performance: associative recall and copying. We find that, while Transformers are robust to optimization hyperparameters, the performance of modern recurrent models suffers from critical instabilities: success is confined to an extremely narrow window of learning rates, outside of which accuracy drastically drops. This issue can confound performance evaluations and expressivity conclusions, revealing a fundamental mismatch in the loss landscape of modern recurrent models compared to Transformers. We demonstrate that this brittle optimization has a direct impact on scaling, causing SSMs to favor width over depth. Indeed, we also find that, while the 1-layer Transformer's performance on recall does not exceed random guessing, well-tuned Mamba and other SSMs can learn to recall with one layer, yet with dynamics that do not resemble the formation of induction heads. Taken together, our findings suggest that a crucial differentiator between these architectures lies not just in their expressivity but in their fundamental learnability properties, pointing to optimization stability as a key challenge for the future of SSMs.

## 1 INTRODUCTION

Since early developments (Rumelhart et al., 1986; Elman, 1990), RNNs have driven progress in machine learning techniques for sequential data, with milestones such as Echo-State Networks (Jaeger, 2001) LSTM (Hochreiter & Schmidhuber, 1997) and GRU (Cho et al., 2014). However, two problems severely limit the application of RNNs in modern times: first, GPU architectures struggle with sequential processing. Secondly, it is widely known that RNNs are hard to train due to vanishing and exploding gradients issues (Bengio et al., 1994; Hochreiter et al., 2001; Pascanu et al., 2013).

**Attention.** These challenges have led to the introduction of a different paradigm: the Attention mechanism, implemented around the Transformer architecture (Vaswani et al., 2017). Instead of processing inputs sequentially while building up internal memory (RNNs), Attention computes pairwise interactions between data points, allowing for modeling direct links between elements in a sequence and thus mitigating vanishing gradients. While Attention, being based on matrix multiplications, is extremely GPU efficient, computing pairwise interactions results in $O(L^2)$ inference and memory complexity, where $L$ denotes the input sequence length. For this reason, techniques such as patching (Dosovitskiy et al., 2021; Pagnoni et al., 2024), gradient checkpointing (Chen et al., 2016), and FlashAttention (Dao et al., 2022; Dao, 2023; Shah et al., 2024) become of paramount importance when training and deploying Attention-based models at scale. Despite this limitation, Transformers successfully power most state-of-the-art architectures we use today: beyond LLMs (Devlin, 2018; Brown et al., 2020; Team et al., 2024), Attention found widespread application in vision (Dosovitskiy et al., 2021; Touvron et al., 2021; Bertasius et al., 2021; Liu et al., 2024a), graph processing (Ma et al., 2023), and genome analysis (Dalla-Torre et al., 2024), among others. Nevertheless, the quadratic complexity of Attention has remained a pressing limitation, prompting numerous efforts to develop more efficient approximations (Wang et al., 2020; Choromanski et al., 2020; Chen et al., 2021; Lee-Thorp et al., 2022). Many of these approaches have even revealed connections to recurrent formulations (Katharopoulos et al., 2020; Schlag et al., 2021).

**SSMs and other linear token mixers.** More recently, we have witnessed a resurgence of RNNs in state-of-the-art industry-size applications such as language modeling (Qwen Team, 2025). Sparked by the S4 model (Gu et al., 2020; 2022), which surpassed Attention-based models on long-range reasoning tasks (Tay et al., 2020), we have rapidly seen in the last year a drastic increase in the usage of RNNs in deep architectures, albeit in a linear[1] form that guarantees both $O(L)$ memory/inference complexity and fast computation on GPUs (Martin & Cundy, 2018; Orvieto et al., 2023) while matching or surpassing Transformers on downstream tasks: prime examples are State-space Models (SSMs) such as Mamba(2) (Gu & Dao, 2024; Dao & Gu, 2024), along with variants based on similar ambitions (De et al., 2024; Peng et al., 2024; Yang et al., 2025). These novel fast recurrent processing strategies sparked the interest of many practitioners in the field, leading to novel applications in several domains, including vision (Liu et al., 2024b; Liang et al., 2024), audio generation (Goel et al., 2022), and reinforcement learning (Lu et al., 2023).

**Conflicting Views on SSMs vs. Transformers.** The resurgence of RNNs has led to a fascinating debate within the community (Gu, 2025). On one hand, theoretical works suggest deep parallels between architectures (Dao & Gu, 2024; Ali et al., 2024). On the other, empirical studies (Waleffe et al., 2024) as well as theoretical expressivity analyses (Arora et al., 2023; 2024; Jelassi et al., 2024) suggest a downstream performance gap [2], indicating that Transformers outperform SSMs on tasks that require strong copying or in-context learning abilities, e.g. MMLU (Hendrycks et al., 2009).

This discrepancy raises a crucial question that motivates our work: is this gap caused by fundamental limitations in what SSMs can express, or by practical challenges in what they can learn during training? To investigate this, inspired by the large-scale investigation of pretrained models by (Waleffe et al., 2024) and in need of a *simple-yet-insightful* small-scale setup to perform thorough ablations at academic scales, we focus on two well-established benchmarks shown to be highly correlated with language modeling retrieval and in-context learning abilities and often studied to assess basic expressivity properties: multi-query associative recall (MQAR) (Arora et al., 2023) and copying (Jelassi et al., 2024). Our empirical analysis encompasses over 3,000 runs and approximately 20,000 GPU hours. While some analyses point to theoretical limitations like the finite hidden state of recurrent models (Jelassi et al., 2024), we hypothesize that conclusions may be confounded or exaggerated by optimization issues. Specifically, we posit that modern SSMs inherit the notoriously difficult training dynamics—such as vanishing and exploding gradients—that were extensively documented in both classical (Pascanu et al., 2013) and modern (Zucchet & Orvieto, 2024) RNNs. Our initial findings, presented in Figure 1, strongly support this learnability-focused perspective.

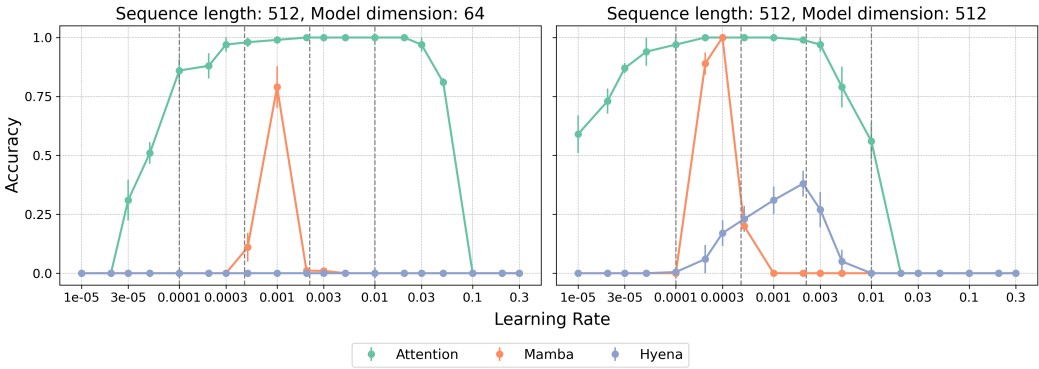

Figure 1: *Performance on MQAR (mean and relative max-min errors using 5 seeds) after an extensive learning rate grid search. Unlike attention, the window of suitable learning rates for Mamba and Hyena is relatively narrow. We compare our grid search with the one used by Arora et al. (2023) (**dashed vertical lines**) to highlight how the suitable learning rate can be missed.*

Figure 1 points to a crucial confounder when comparing SSM and attention capabilities: while fundamental expressivity issues exist between such model classes, the main driver of poor performance

---

[1]Modern RNNs such as State-space Models are linear in the hidden-to-hidden state interactions, but have recurrent formulation that is non-linear in the input, see (Cirone et al., 2024)

[2]Training perplexity can be instead similar, as reported in Gu & Dao (2024).

can be unsuccessful optimization. This leads us to our central thesis: ***Transformers differ from SSMs** not in terms of expressive power but mainly because of their optimization dynamics.*

Building on this insight, in this paper we take a closer look at this learnability gap.

- **Critical Optimization Instability**: We demonstrate that on both associative recall and copying tasks, the success of modern recurrent models is confined to an extremely narrow window of learning rates. This reveals a critical instability not present in Transformers, suggesting that prior empirical expressivity comparisons may have been confounded by suboptimal tuning.
- **Contrasting Scaling Behavior**: We reveal opposing model scaling strategies for Transformers and SSMs. Consistent with prior research, recurrent models benefit most from increased width, as relying on a larger hidden state facilitates information retrieval as the sequence length increases (Orvieto et al., 2024; Gu, 2025). Indeed, while a single-layer Mamba (properly tuned) can solve recall, single-layer attention model fails to solve the task, while a two-layer version succeeds. This suggests that further research should avoid theoretical one-layer comparisons.
- **Divergent Single-Layer Dynamics:** We analyze the training dynamics of single-layer models, finding that a 1-layer Transformer also exhibits a loss drop reminiscent of induction head formation ( (Olsson et al., 2022; Bietti et al., 2023)), while failing to fit the training set. Meanwhile, recurrent models show smoother training dynamics in most setups, with no clear evidence for the formation of induction heads. This finding points to severe mismatches in the landscape geometry.
- **Architectural Drivers to Stability**: Through targeted ablations, we show that the single-layer performance of Mamba is critically dependent on its 1D convolution while, conversely, adding a simple convolution to the single-layer Transformer enables it to solve MQAR. We also study how newer SSMs, such as DeltaNet (Yang et al., 2024), can improve optimization stability in MQAR.

## 2 BACKGROUND AND RELATED WORKS

**Associative Recall.** With the rise of foundation models, deep learning has made significant advances, sparking growing interest in evaluating their reasoning capabilities. One key aspect of reasoning is the ability to recall previously encountered information. Intuitively, given the input

> *"**Hakuna Matata** means **no worries** for the rest of your days.*
> *"**Hakuna Matata** means ..."*

a well-performing model should predict ***"no worries"*** with high likelihood. Building on this idea, the synthetic associative recall (AR) task, introduced by (Olsson et al., 2022), gained popularity as an efficient reasoning benchmark to assess promising architectures at a relatively low cost. The task is structured as follows: Each sample consists of a sequence of tokens sampled from a fixed vocabulary $V$, representing alternating key-value pairs. Given such a sequence and a key that appeared earlier, the model must correctly infer its corresponding value: For instance, given the sequence:

$$A \quad 6 \quad I \quad 9 \quad C \quad 7 \quad P \quad 1 \quad S \quad 4 \quad D \quad 2$$

and the key $C \rightarrow$ ? the model should predict 7.

A crucial aspect is that the tokens serve interchangeably as keys and values among samples—they are drawn from the same vocabulary rather than separate sets. Consequently, the model cannot rely on preassigned roles for tokens. Moreover, since roles and positions vary across data points, the model cannot memorize a fixed mapping but must instead infer the correct associations in-context.

**Multi-Query Associative Recall.** Building on previous research (Arora et al., 2023), our experiments employ a variation of AR known as multi-query associative recall (MQAR). This choice is motivated by the fact that standard AR is typically used to evaluate the ability of recurrent models to capture long-range dependencies using extremely long sequences—an area where Attention-based models often struggle due to memory constraints. However, at the scale of our experiments, MQAR presents a more challenging and relevant task even with relatively small sequences. There are two key distinctions between MQAR and its standard counterpart, both of which align more closely with the characteristics of natural language. First, it introduces a significantly larger vocabulary: from the 50 tokens of standard AR to approximately $8,000$ tokens in MQAR. This makes the task more representative of real-world language processing where the vocabulary size is on the order of hundreds of thousands of words. Second, instead of recalling a single key-value pair, MQAR requires the

model to retrieve multiple values based on multiple queries. This more accurately mirrors the nature of language, where meaning is often derived from groups of words and interrelated concepts rather than isolated tokens. For instance, given the same sequence $A$ $6$ $I$ $9$ $C$ $7$ $P$ $1$ $S$ $4$ $D$ $2$ and multiple keys $C \rightarrow ?$ $A \rightarrow ?$ $D \rightarrow ?$ we ask the model to recall the relative values 7, 6 and 2. Notably, if we were to restrict the model to retrieving only one key-value pair, the task would reduce to AR. Prior studies have demonstrated that this variant highlights more effectively the differences between Attention-based and recurrent models.

**Copying.** Another fundamental benchmark used to evaluate sequence models' memory and recall capabilities is the copying task (Jelassi et al., 2024). Differently from MQAR, the model's objective is to accurately retrieve the whole (not selectively) initial string of tokens in place after a delimiter.

**Induction heads.** While investigating the capabilities of Transformers in few-shot learning, previous works (Olsson et al., 2022; Bietti et al., 2023) showed that during training, with Transformers with at least 2 layers, a special kind of attention heads called "induction heads" is formed, causing a noticeable drop in the training loss, while giving a sudden boost in in-context learning performance. More formally, induction heads are implemented by a circuit consisting of a pair of Attention heads in different layers that work together to copy or complete patterns. The first Attention head copies information from the previous token into every other tokens, making it possible for the second Attention head to attend to tokens based on what happened before them, rather than their own content. Specifically, the second head (the proper "induction head") searches for a previous place in the sequence where the present token **A** occurred and attends to the next token (call it **B**), copying it and causing the model to be more likely to output **B** as the next token. That is, the two heads working together cause the sequence ...[**A**][**B**]...[**A**] to be more likely completed with [**B**].

Induction heads are named by analogy to inductive reasoning, where we might infer that if **A** is followed by **B** earlier in the context, **A** is more likely to be followed by **B** again later in the same context. Induction heads are capable of crystallizing that inference. They search the context for previous instances of the present token, attend to the token which would come next in the repeated pattern, and increase its corresponding logit value. Induction heads attend to tokens that would be predicted by basic induction (over the context, rather than over the training data).

**Transformers and SSMs.** Let $X \in \mathbb{R}^{N \times d}$ a generic input consisting of $N$ elements in $d$ dimensions. Basic state-space models (SSMs) (Gu & Dao, 2024) compute outputs via a recurrence:

$$Z_i = A_i Z_{i-1} + B_i X_i$$
$$Y_i = C_i Z_i + D_i X_i,$$

where $Z_0 = 0$ and $A_i, B_i, C_i, D_i$ are input-dependent matrices. In the S6 block (Gu & Dao, 2024), they are parametrized as functions of $X_i$, yielding a structured recurrence.

This system admits a an attention formulation (Sieber et al., 2024; Dao & Gu, 2024): $Y = \Phi_{\text{S6}}^X \cdot X$,

$$\Phi_{\text{S6}}^X = \begin{pmatrix} C_0 B_0 + D_0 & 0 & \cdots & 0 \\ C_1 A_1 B_0 & C_1 B_1 + D_1 & \cdots & 0 \\ \vdots & \ddots & \ddots & \vdots \\ C_N \prod_{k=1}^N A_k B_0 & \cdots & C_N A_N B_{N-1} & C_N B_N + D_N \end{pmatrix}. \tag{1}$$

Mamba2 (Dao & Gu, 2024) and (Gated) DeltaNet (Yang et al., 2024; 2025) also share this view, yet their efficient formulation introduces further state expansion and parameter sharing options.

## 3   CLOSER LOOK INTO AR PERFORMANCE

Building on previous research, we provide an in-depth analysis of the differences between attention-based and recurrent models through the lens of AR. Prior studies (Arora et al., 2023) have shown that Transformers are inherently well-suited for solving the MQAR task, achieving perfect accuracy across all settings. In contrast, it was argued (both theoretically and empirically) that new recurrent models can only solve MQAR if the hidden dimension is roughly equal to the sequence length. However, a key aspect that has been **overlooked** is the crucial role of optimization in recurrent models, particularly the use of an effective grid search for the choice of learning rate.

**The memory bottleneck hypothesis.** Recurrent models update their hidden state (which serves as a compressed representation of past information) at each time step, using the current input. Since

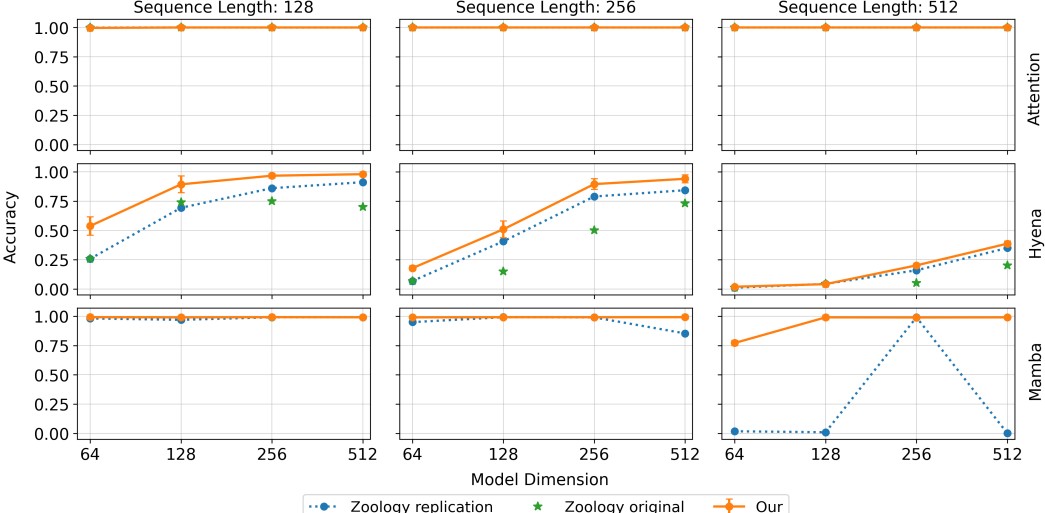

Figure 2: *Performance of 2-layers models in MQAR. We report the official results[4] (green stars) and the replication running the original code of (Arora et al., 2023) (dotted blue line). While for replication, we used the learning rate grid by Arora et al. (2023), we note here that, due to high sensitivity to the learning rate (Fig, 1), tuning drastically affects performance. In solid orange, we provide results with a finer grid (cf. Fig.1). Careful tuning of the learning rate gives a general improvement in the performance of recurrent models. This becomes especially crucial in Mamba, where the task becomes solvable at high sequence lengths >> hidden size. The results show the mean and relative max-min errors for 5 seeds. Attention always solves the task (all curves overlap).*

the model only has access to its hidden state and the current input, its ability to recall previous information depends on how effectively it compresses past data into this state. For instance, with a simplified analysis assuming uniform distribution over strings, Jelassi et al. (2024) showed that to successfully copy input strings, the hidden size needed grows linearly with the sequence length. In contrast, Transformers dynamically access all previously seen inputs through the softmax attention mechanism, allowing for the explicit computation of interactions between tokens. This makes the task of recalling already seen tokens essentially a lookup table problem when two layers work simultaneously, as described by Jelassi et al. (2024); Olsson et al. (2022); Bietti et al. (2023).

**Results.** Compared to previous work, in our experiments, we carefully tuned the learning rates, drastically improving the reported performance for recurrent models (see Fig. 2&1). As shown in Figure 2 (full tables in Appendix A.3 with more models), a finer grid not only enhances average performance across all settings but is also particularly crucial for the Mamba (Gu & Dao, 2024) model. With a more suitable learning rate, Mamba, which was previously shown to struggle with long sequence lengths, becomes capable of solving MQAR at relatively small hidden model sizes. All experimental details for this and subsequent experiments are in Appendix A.2. This highlights a key takeaway for MQAR: the choice of learning rate (and optimization strategy in general) can be decisive in assessing whether a recurrent model can solve the task at all. In the case of Mamba, optimization choices become a discriminative factor, emphasizing the necessity of careful hyperparameter tuning in recurrent models, and further research for improving their high sensitivity.

To further emphasize the critical role of learning rate selection in training recurrent models, we compare performance with respect of our grid search. Figure 1 (and Appendix A.6 with deeper networks) illustrates that Attention-based models maintain strong performance across a relatively wide range of learning rates. In contrast, Hyena and Mamba exhibit a different behavior: performance remains near zero for most learning rates but suddenly reaches near-optimal levels at specific values that may not be included in the grid by Arora et al. (2023). These findings highlight a key distinction between Attention-based and recurrent models: a sparse learning rate grid search can disproportionately impact their training outcomes. **This discrepancy can lead to misleading conclusions** about the capabilities of these models, emphasizing the need for careful tuning.

---

[4]Mamba was not included in the official work but some experiments are documented in the blog post.

## 4 EFFECTS OF WIDTH/DEPTH SCALING INTO AR PERFORMANCE

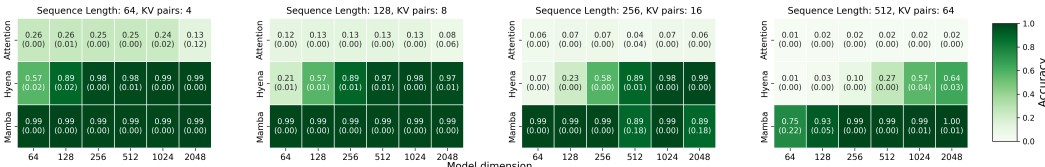

Figure 3: *Performance of 1-layer models on MQAR. We show how for recurrent models, scaling the width boosts performance. On the contrary, Attention models can no longer solve the task anymore as they do in the 2-layer setting, and performances are unaffected by the scaling in width. The results show the mean and relative max-min errors after 5 runs with different seeds.*

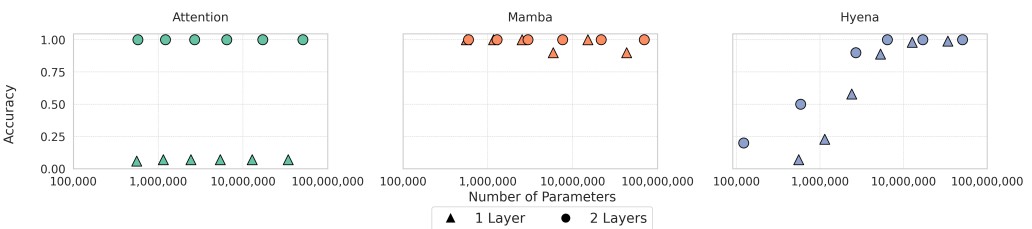

Figure 4: *Scaling behavior (Seq len: 256, KV pairs: 64). Symbols with the same shape and color represent models of increasing dimension from 64 to 2048. We show that the scaling strategy, rather than the total number of parameters, is what primarily impacts performance. Specifically, recurrent models benefit from scaling in width, while attention-based models benefit from scaling in depth.*

While our findings in Sec. 3 show that some recurrent models can exhibit improved performance on MQAR with proper learning rate tuning, we confirm that a sizable gap with Transformers can still be observed at low widths (e.g. Hyena). The experiments of Sec. 3 focused on comparisons of 2-layer architectures, at different sequence lengths and model widths. This choice stems from prior research (Olsson et al., 2022), where Transformers have shown peculiar in-context learning capabilities related to the formation of induction head circuits in 2-layer models. Indeed increasing the number of layers to more than 2 does not provide any further improvement in MQAR performance. With the intention of going *beyond the setup that is known to show strengths for softmax attention*, our objective in this section is to explore the effects of scaling in different configurations.

To achieve this, we conducted experiments analogous to Sec. 3 using single-layer models[5]. By doing so, we aim to decouple the effects of inter-communication between layers and to isolate the impact of each model's fundamental structure. Beyond this, our motivation also comes from the notable connections that have been drawn between Attention and recurrent models (Dao & Gu, 2024; Ali et al., 2024; Sieber et al., 2024; Huang et al., 2025) and on the capabilities of Transformers (Sanford et al., 2024), all of which concern 1-layer models. Our results, presented in Figure 3 (full table in Appendix A.4), reveal two key insights:

1. First, for a fixed sequence length, recurrent models always benefit from scaling in width, as was happening in 2 layers (Sec. 3). That is, expanding the hidden state dimension enhances their performance. This result aligns well with current literature (Jelassi et al., 2024; Orvieto et al., 2024): as already mentioned, at each time step recurrent models store compressed inputs into a hidden state, which serves as a condensed representation of all past information. A larger hidden dimension facilitates less aggressive compression, allowing the model to retain more information.
2. Attention exhibit a surprisingly different behavior: when constrained to a single layer, they fail to solve the task and increasing the hidden dimension does not affect their performance. This is in stark contrast to their strong results in 2-layer architectures, where even the smallest model was sufficient to solve the task in the hardest setting. Interestingly, in this setting Transformers are capable on average of recalling one key-value pair in every setting, suggesting a memory size issue when only one layer is present as also suggested in previous work (Sanford et al., 2024).

---

[5]In this context, a single layer refers to a sequence mixer followed by an MLP.

Our findings highlight a key takeaway from our study: Attention and recurrent models exhibit opposite scaling behaviors in width and depth. In other words, as shown in Fig. 4, rather than the number of parameters, it is the way these models are scaled that has most impact on their performance.

# 5 COPY TASK

To validate that optimization instabilities and scaling behaviors observed on MQAR are not dataset-specific, we conducted a parallel investigation on the copying task studied by (Jelassi et al., 2024).

**Learning Rate Instability.** Just as with MQAR, optimization stability offers a new perspective on the conclusions of (Jelassi et al., 2024). As shown in Figure 5, the Transformer solves the copying task robustly across a wide range of learning rates. In contrast, Mamba's success is again confined to a narrow window.

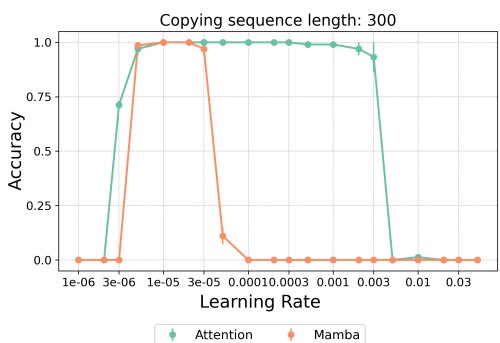

**Parameter Scaling.** We find that attempts to provide fair comparisons by matching parameter counts through increased depth in SSMs are misguided. As shown in Table 1, a deeper but narrower Mamba fails to copy, whereas a shallower but wider Mamba with the same parameter count succeeds. This reinforces our claim that architectures must be scaled along their preferred axes, width for SSMs and depth for Transformers, to unlock their potential.

Figure 5: *Performance of a Transformer with RoPE and Mamba on the copy task following (Jelassi et al., 2024) implementation. This task also highlights the narrow window of suitable learning rates that allows Mamba to solve the task.*

Table 1: *Performance on the copy task. When comparing Transformers and SSMs with the same number of parameters, it is crucial to scale the latter in width rather than depth.*

| Architecture | # Layers | Width | # Parameters (M) | Accuracy (%) |
|---|---|---|---|---|
| Attention (RoPE) | 12 | 1024 | 150 | 100% |
| Mamba | 12 | 1024 | 80 | 0% |
| Mamba | 24 | 1024 | 150 | 16% |
| Mamba | 12 | 1408 | 150 | 100% |

# 6 1-LAYER TRAINING DYNAMICS AND INDUCTION HEADS PHENOMENON

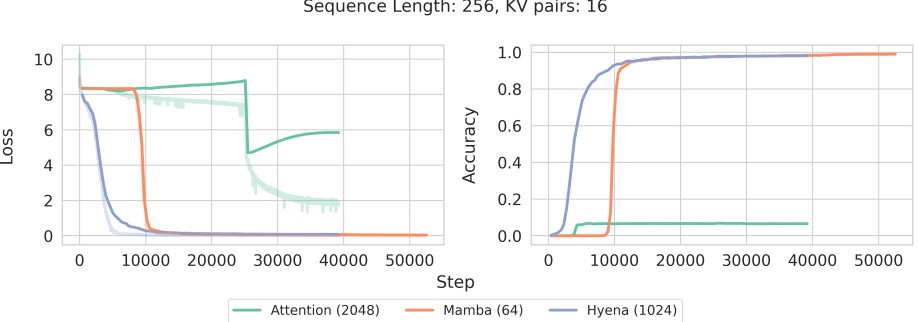

Figure 6: *Training (lower opacity) and validation dynamics of 1-layer models in MQAR. We report within brackets the smallest width that solves the task, if possible; or otherwise the largest width we tried (for Attention). Differently from Mamba, Hyena requires the model dimension to exceed the sequence length. Both exhibit smooth learning dynamics, leading to perfect performance. Attention shows a loss bump, but without accuracy gains, suggesting an attempt to form induction heads that a single-layer Transformer fails to leverage effectively.*

Sec. 4 and 5 sparked our curiosity, leading us to explore the single-layer architecture setup further in MQAR, to understand why Attention hits a performance ceiling while recurrent models can solve the task. This analysis is especially intriguing given the strong connections that have been proposed between attention and Mamba in Ali et al. (2024); Dao & Gu (2024); Sieber et al. (2024).

In this section, we analyze the training dynamics of Hyena, Attention and Mamba models. As illustrated in Fig. 6 we identify two main patterns. First, Hyena exhibits consistently smooth learning dynamics, with a gradual and steady improvement that eventually leads to convergence at the solution. Specifically, loss reductions align closely with increases in accuracy. Differently, Attention accuracy remains largely unchanged throughout training. A similar trend appears in the test loss, which remains relatively stable until a sudden bump occurs, after which the test loss settles again. This bump resembles the formation of an induction head circuit (Olsson et al., 2022), and to the best of our knowledge has previously only been observed during the training of multi-layer transformer architectures. However, as opposed to the 2-layer models, this phase transition in the loss does not correspond to an accuracy improvement for attention. Based on previous work (Olsson et al., 2022), we hypothesize that during this phase transition, the Attention mechanism *attempts* to form induction heads. However a single-layer transformer lacks the expressivity needed to effectively leverage this mechanism for task resolution. Interestingly, the dynamic of Mamba is mixed:

1. Like single-layer Attention models, we report a significant loss bump, reinforcing the connection between Mamba and Attention mechanisms, as suggested in Ali et al. (2024); Dao & Gu (2024).
2. However, unlike transformers, Mamba can successfully solve the task even in a single-layer setting, provided the learning rate is properly tuned, similarly to other recurrent models.

Our results highlight how Attention and recurrent models share some common ground, yet distinct inductive biases. Moreover, their performance strongly interacts with the optimization algorithm at hand (in our case, Adam (Kingma, 2014)), as we also saw in Fig. 1. Understanding these nuances is key to optimally leverage both architectures, towards hybrid models (Waleffe et al., 2024).

## 7 ARCHITECTURAL DRIVERS OF PERFORMANCE AND STABILITY

Our results so far highlight key differences between Transformers and SSMs. Notably, while Mamba demonstrates greater expressivity, successfully solving the task even in a single-layer setting, it presents optimization challenges in terms of learning rate stability. In contrast, Transformers exhibit remarkable stability across a wide range of learning rates during training in the 2-layer setting. To address this discrepancy, we conduct a series of ablation studies aimed at:

1. aligning the backbone of both models (full details in Appendix A.1) and identify the source of Mamba's superior performance in 1-layer, summarized in Table 2 and Appendix 3;
2. exploring new architectural variants that promote more stable training dynamics.

**Convolutions.** Inspired by (Li et al., 2024), we begin by incorporating a 1D convolution before the QKV matrix projections to bring in locality, enabling the model to solve MQAR with just one layer. These observations suggest the 1D convolution is a critical component for enabling expressivity in shallow sequence models. Indeed, while the original 2-layer Mamba is robust to convolution removal, removing it from a 1-layer Mamba reduces its accuracy to the same failure point as the 1-layer Trans-

Table 2: *MQAR performance of 1-layer Attention and Mamba with ablations on architecture.*

| Model | Accuracy |
|---|---|
| Attention | 2% |
| Attention + Conv on QKV | 99% |
| Mamba | 99% |
| Mamba w\o conv1d | 2% |
| Mamba w\o gating | 98% |
| S6 + MLP (as a Transformer) | 98% |

former (Arora et al., 2025). This new finding provides a strong mechanistic link: in terms of raw expressivity, a 1-layer Mamba without convolution performs approximately identically to a 1-layer Transformer. However, the narrow learning rate window remains a persistent property of the SSM.

**Backbone ablation.** We further modify the Mamba architecture by: (1) removing the gating mechanism, and (2) replacing the standard Mamba block with the individual sequence mixer S6, followed by an MLP, mirroring the Transformer's architecture. Despite these alterations, Mamba performs well when properly tuned, suggesting the sequence mixer (S6) is at the root of its expressivity.

**Positional Encodings.** We tested whether adding various Positional Embedding (PE) strategies could improve SSM performance and stability. Our findings reported in Table 4 show that PE has a negligible impact on performance. This result reinforces that the core recurrent structure is the dominant mechanism for processing sequence order in these models.

**Newer architectures.** To better understand what contributes to training stability, we also evaluate architectural variants designed for improving Mamba and solve the MQAR task. In particular, we test Mamba2 (Dao & Gu, 2024) and DeltaNet (Yang et al., 2024) as shown in Figure 7. While the performance of Mamba2 is slightly more stable, Transformer-level robustness is only achieved by DeltaNet. A closer look at the DeltaNet update rule reveals that its mixing is based on Householder matrices. As such, the off-diagonal terms such as $C_N \prod_{k=1}^{N} A_k B_0$ do not necessarily incur in vanishing gradients. Instead, in both Mamba and Mamba2, $A_k$ includes a decay rate that induces vanishing gradients and fast decay of off-diagonal terms, as recently pointed out by Trockman et al. (2024). We hypothesize this is the main distinction unlocking stable optimization in DeltaNet.

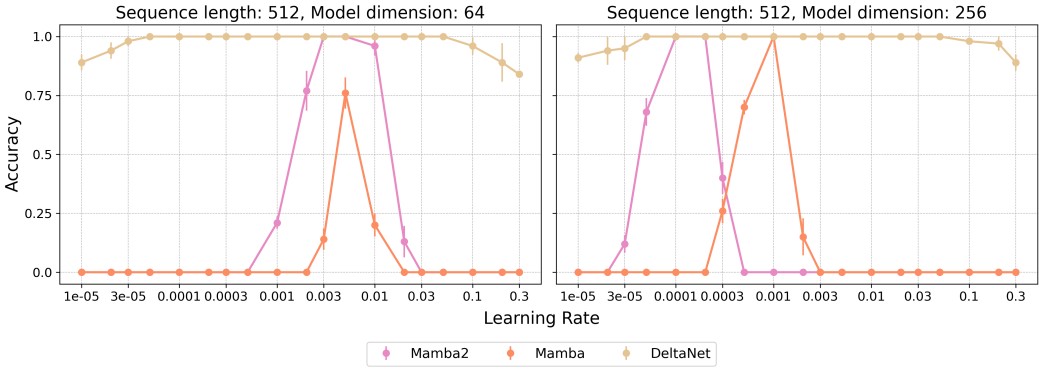

Figure 7: *Performance of newer 1-layer models on MQAR. Here we show how having a larger hidden state marginally helps stability, as in Mamba2 and especially in DeltaNet. Our results are presented for model dimensions up to 256, which was the maximum size supported by the DeltaNet implementation. The results show the mean and relative max-min errors using 5 seeds.*

## 8 DISCUSSION AND CONCLUSIONS

In this work, we dissected the practical differences between Transformers and modern recurrent models on the associative recall and copying tasks. Our findings demonstrate that a crucial differentiator lies not just in their theoretical expressivity, but in their fundamental learnability. We demonstrated that modern SSMs exhibit a critical optimization instability, with success confined to a narrow learning rate window—a finding that re-contextualizes prior performance evaluations. Additionally, we observed contrasting scaling behaviors: recurrent models benefit from increased width, whereas Transformers struggle in a single-layer configuration. Interestingly, despite their poor performance, single-layer Transformers exhibit training dynamics resembling the induction head phenomenon, previously reported only in multi-layer settings. Instead, Mamba displays similar behavior but successfully solves the task. Finally, our ablations show how the convolution makes Mamba mechanistically similar to a Transofrmer. More recent architectures, like DeltaNet, can enhance performance and stability. The central implication of our work is that future research on efficient sequence models should treat optimization stability as a first-class objective, alongside expressivity and computational cost. While our findings are compelling, we acknowledge that our analysis is conducted on synthetic benchmarks highly correlated with in-context learning. Validating these dynamics on downstream language modeling tasks is a critical next step. Furthermore, a formal theoretical explanation for the optimization brittleness we empirically observe remains an important open question. Looking ahead, by showing that modern recurrent models can be as expressive as Transformers on these tasks but are harder to train, our work underscores the importance of learnability in the path towards understanding and building the next generation of sequence models.

## ETHICS STATEMENT

As our research focuses on a foundational analysis of sequence models using synthetic data we do not foresee any direct ethical concerns arising from our methods or findings.

## REPRODUCIBILITY STATEMENT

To ensure the full reproducibility of our experimental findings, all of our code is made publicly available. Our implementation builds upon the open-source codebases provided by (Arora et al., 2023) and (Jelassi et al., 2024). All specific hyperparameters, architectural details, and training configurations for each model and task are comprehensively documented in Appendix A.2. We believe these measures provide everything necessary for the direct replication of our results.

## ACKNOWLEDGMENT OF AI-ASSISTED TOOLS

AI-assisted editing tools were used to check grammar.

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

# A APPENDIX

## A.1 MODIFIED ARCHITECTURES

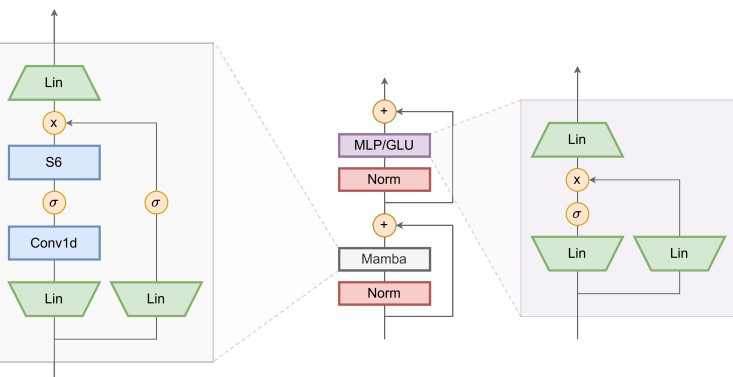

Figure 8: *Architecture of Mamba following the architecture of a Transformer given by sequence mixers interleaved by MLPs*

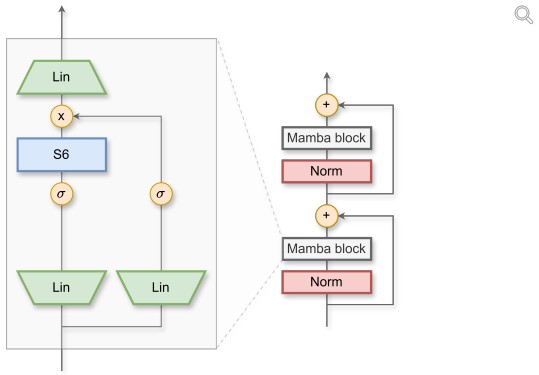

Figure 9: *Architecture of Mamba without the conv1d*

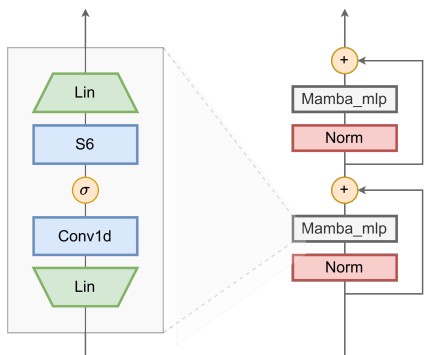

Figure 10: *Architecture of Mamba without the gate*

## A.2 EXPERIMENTAL DETAILS

In this section, we describe the experimental setup used throughout our study. Clearly outlining these details is crucial for interpreting the results presented in subsequent sections. Our implementation is inspired by methodologies from Zoology (Arora et al., 2023) while the relative codebases of original models can be found in the zoology github page https://github.com/HazyResearch/zoology/tree/main.

**Data.** The dataset consists of sequences of tokens representing key-value pairs. Tokens are sampled from a fixed vocabulary of $8,192$ tokens. Within each sequence, key-value tokens are assigned randomly, ensuring that the model cannot learn a static mapping. Consequently, each sample is independent, requiring the model to infer the role of tokens in context rather than relying on memorization. The synthetic dataset is structured with four specific sequence lengths, each paired with a corresponding number of key-value pairs to recall:

- 64 tokens with 4 key-value pairs;

- 128 tokens with 8 key-value pairs;

- 256 tokens with 16 key-value pairs;

- 512 tokens with 64 key-value pairs.

For the first three sequence lengths, the ratio of key-value pairs to sequence length is $1 : 16$, whereas for the longest sequence, the ratio is $1 : 8$, making it the most challenging case. For each sequence length, a dedicated dataset is created, consisting of $100,000$ training samples and $3,000$ test samples. Model evaluation is performed by training each model on a specific sequence length and subsequently assessing its performance on that same length.

**Models** Our experiments utilize a total of six main models + others used in the ablation studies:

- Two Attention-based models: Attention and Based.

- Four recurrent models: H3, Hyena, RWKV and Mamba.

- Other Ablations such as Attention + Convolution, Mamba without specific components etc.

Each model is tested across six model dimensions: 64, 128, 256, 512, 1024 and 2048. Additionally, models are implemented in two configurations: 1-layer and 2-layer. Notably, a "layer" in our context refers to the concatenation of two blocks: a sequence mixer (e.g., attention, RWKV, etc.) followed by an MLP. Thus, a 1-layer model consists of two blocks, aligning with the terminology used in prior work (Arora et al., 2023; Olsson et al., 2022). Positional information is used only in Attention and Based.

**Training and Evaluation** We used GPU A100 with 80GB of memory in all our experiments. We trained for 50 epochs using AdamW as optimizer, weight decay 0.1, warmup duration 10%, linear warmup. All the experiments took between 10 minutes and 18 hours based on the model architecture, the model dimension and the sequence length. The batch size varied depending on the sequence length: 128 for sequence length 512, 256 for sequence length 256 and 512 otherwise. Each configuration (combining model type, model dimension, and sequence length) undergoes a learning rate sweep to identify the optimal learning rate. The reported accuracy for each configuration corresponds to the best performance achieved across the tested learning rates. We want to highlight that the accuracy reported should be interpreted as the average percentage of key-value pairs correctly labeled. Specifically, achieving $50\%$ accuracy with sequence length 64 and 4 as relative number of key-value pairs means that on average the model recalls correctly 2 values given 4 keys. To ensure robustness, all experiments are conducted using five random seeds (42, 123, 777, 456 and 789), with results reported as the mean and standard deviation across these trials.

## A.3 MQAR PERFORMANCE OF 2-LAYER MODELS

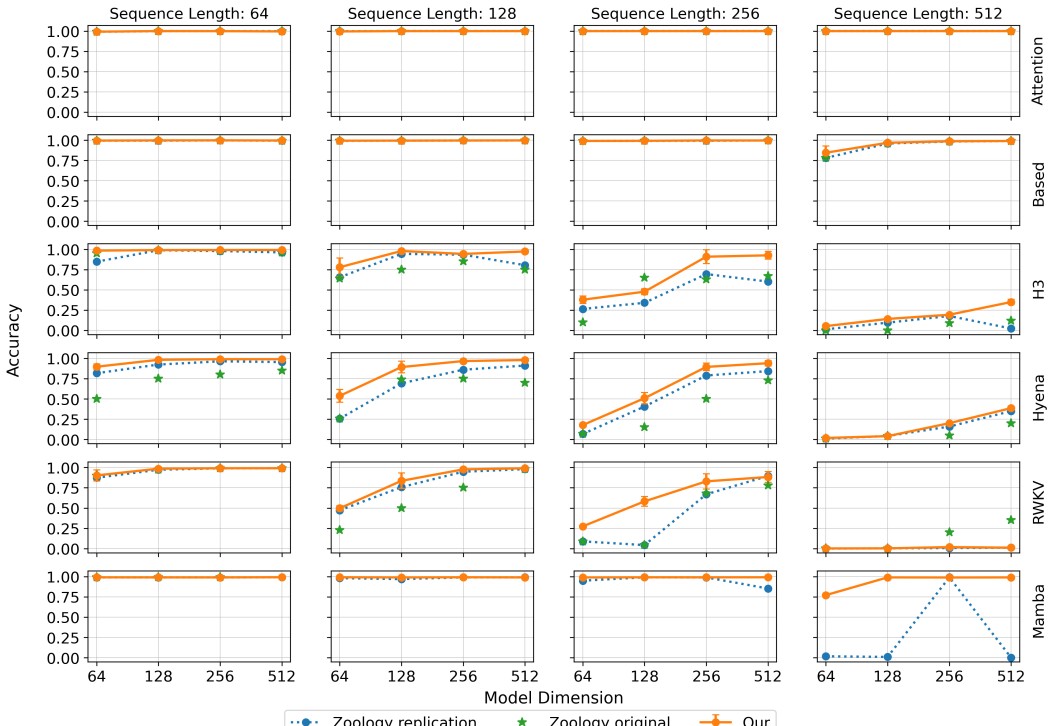

Figure 11: *Performance of 2-layers models. We report the official results[7] (green stars) and the replication running the original code of (Arora et al., 2023) (dotted blu line). While for replication, we used the learning rates grid by Arora et al. (2023), we note here that, due to high sensitivity to the learning rate (Fig, 1), tuning drastically affects performance. In solid orange, we provide results with a finer grid (cf. Fig.1). Careful tuning of the learning rate gives a general improvement in the performance of recurrent models. This becomes especially crucial in Mamba, where the task becomes solvable at high sequence lengths >> hidden size. The results show the mean and relative max-min errors for 5 seeds. Attention always solves the task (all curves overlap).*

## A.4 MQAR PERFORMANCE OF 1-LAYER MODELS

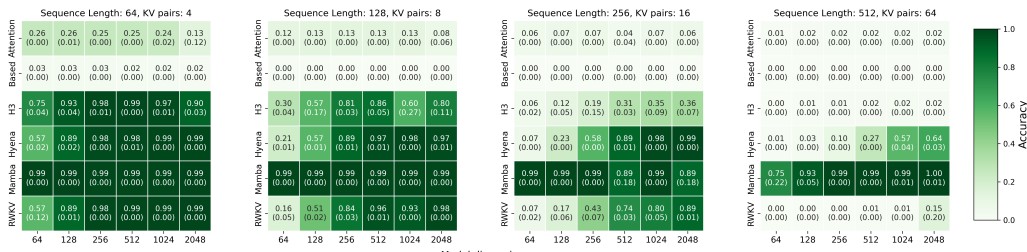

Figure 12: *Performance of 1-layer models on MQAR. We show how for recurrent models, scaling the width boosts performances. On the contrary, Attention models cannot solve the task anymore as in the 2-layer setting, and performances are unaffected by the scaling in width. The results show the mean and relative max-min errors after 5 runs with different seeds.*

---

[7]Mamba was not included in the official work but some experiments, with different settings compared to ours, are documented in the blog post.

## A.5 MQAR PERFORMANCE OF ABLATED 1-LAYER MODELS

Table 3: *Performance of ablated 1-layer architectures on MQAR. Convolutions allows perfect accuracy for 1-layer Mamba and Attention. Interestingly, we observe that applying the convolution to either the Key or Value matrix alone is sufficient to achieve the same performance gains.*

| Model | Solves MQAR |
|---|---|
| Attention | 2% |
| Attention + Conv on QKV | 99% |
| Attention + Conv on K | 99% |
| Attention + Conv on V | 99% |
| Attention + Conv on Q | 2% |
| Mamba | 99% |
| Mamba w\o conv1d | 2% |
| Mamba w\o gating | 98% |
| S6 + MLP (Mamba as a Transformer) | 98% |

Table 4: *Effect of various PEs in MQAR. Performance are comparable and aligns with the fact that recurrent model already encode positional information in their recurrence*

| Architecture | No PE | Absolute PE | Relative PE | Learned PE |
|---|---|---|---|---|
| Mamba | $0.99 \pm 0.01$ | $0.99 \pm 0.01$ | $0.99 \pm 0.01$ | $0.99 \pm 0.01$ |
| Hyena | $0.29 \pm 0.9$ | $0.34 \pm 0.10$ | $0.32 \pm 0.06$ | $0.30 \pm 0.07$ |
| RWKV | $0.22 \pm 0.08$ | $0.21 \pm 0.12$ | $0.24 \pm 0.12$ | $0.22 \pm 0.14$ |
| H3 | $0.25 \pm 0.12$ | $0.26 \pm 0.10$ | $0.27 \pm 0.07$ | $0.30 \pm 0.16$ |

## A.6 MQAR PERFORMANCE OF 12-LAYER MODELS

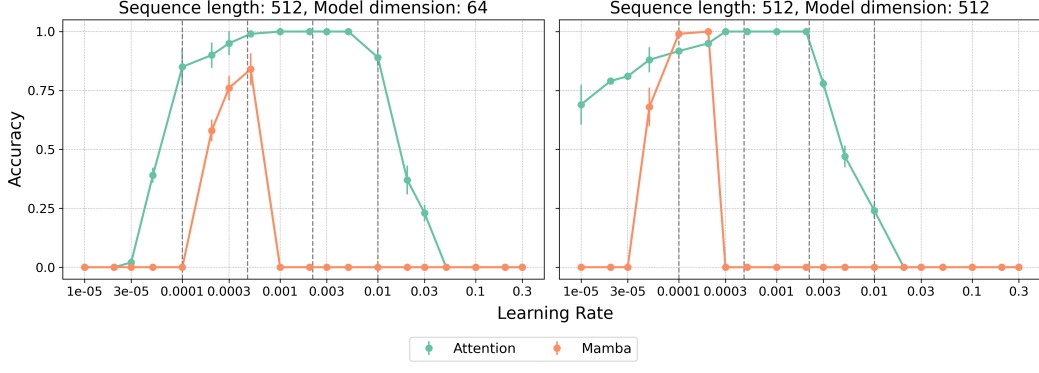

Figure 13: *Performance of 12-layers models. We show how the window of learning rates is related to the specific sequence mixer (Mamba) rather than the number of layers (12 in this case). In fact, performances actually saturates already with just 2 layers given the nature of the task. The results show the mean and relative max-min errors for 5 seeds.*

