# OpenReview forum: "Revisiting Associative Recall in Modern Recurrent Models"
_ICLR.cc/2026/Conference — Submitted to ICLR 2026_

### Official Review · Reviewer_3xjD · 2025-10-14

**Soundness:** 2
**Presentation:** 3
**Contribution:** 1
**Rating:** 2
**Confidence:** 4

**Summary:**

This work provides a study that focuses on the Associative Recall (AR) task, which requires the model to dynamically infer the correct value associated with a given key from a sequence provided as context. The authors conduct a number of different analyses on models such as transformers and state space models, showing that different hyper-parameter choices can affect different models in very different manners. For example, learning rate and model width have surprisingly different effects. Overall the results of this work provides evidence that such a task is very sensitive to to hyper-parameter choices and therefore more careful consideration needs to be placed onto using it as a standardized benchmark for language models.

**Strengths:**

The authors produce a comprehensive empirical study on the associative recall task and reveal some potentially interesting features that could be useful to consider when designing models to work on this task.

**Weaknesses:**

My main concern is the lack of direct explanation that accompanies the results that are provided by the authors; while it is appreciated that there are findings that are more interesting and not well explored in literature (example the number of layers required by some linear recurrent models to learn associative recall tasks), there's ultimately no real explanation for why this is being observed. As a result, the results can appear to be ablating on a number of different variables while lacking a main finding that summarizes the effort as a whole.

Furthermore, some of the results the authors provide such as on the copying task or on the induction heads task with 1-layer are not significantly novel; perhaps it is a positioning issue that requires re-writing but in its current form it does appear to be simply ablating on prior work without a large amount of novel findings.

**Questions:**

See above; it would be appreciated if the authors could provide a more comprehensive theory as to what is being learned by the model and why the different observations occur. Given the synthetic nature of the task, I believe that the authors should focus more effort on this front.

---

> ### Author Response · Authors · 2025-11-24
> **Official comment by Authors**
>
> We thank the Reviewer for the constructive and insightful feedback. We genuinely appreciate your recognizing that our findings reveal "potentially interesting features" for model design, which is precisely the motivation behind our empirical analysis. We found your questions valuable and pertinent and have enabled us to refine our narrative and with the following we would like to give a clear and unified thesis.
>
> ----------
>
> 1. Optimization Instability as the Core Differentiator
>
> Our paper's central thesis is to provide the empirical diagnosis: the fundamental difference between modern SSMs and Transformers lies not in their **expressivity** (what they can represent), but in their **learnability** (what they can practically learn during training).
> - Optimization instability as the root cause: The central finding is that modern SSMs suffer from critical optimization instability, with success confined to an extremely narrow learning rate window (Fig. 1). This is a profound, persistent challenge inherited from classical RNNs and is the main confounder skewing prior performance comparisons.
> - The Unified Explanation: This "learnability" lens is the single, unifying explanation for all our observations:
> It explains the failure of SSMs in prior work, as suboptimal tuning can easily miss the narrow window. It inspired us in evaluating other differences when using these architectures to solve associative recall, leading to the contrasting scaling behavior.  SSMs are forced to prioritize **width** to ease the information compression task, while Transformers leverage **depth**.
>
> -------
>
> 2. Novel Mechanistic Insights and Resolution of Novelty
>
> We understand why the novelty of our 1-layer findings may be questioned, particularly given the extensive work on mechanistic interpretability. To clarify, our contribution here is to contextualize the phenomena observed in the 1-layer setting against the established mechanisms of deeper models. Prior foundational research on Transformer mechanisms, notably by Olsson et al. (2022), Bietti et al. (2023) and subsequent work, established that the formation of the induction head circuit, a key mechanism for in-context learning, requires a **minimum of two attention layers** to function effectively.
>
> - 1-Layer Transformer Dynamics: Our novel finding is that the 1-layer Transformer, despite its inability to solve the task (achieving only 0–2% accuracy), still exhibits the characteristic loss bump during training (Fig. 6). We hypothesize that this bump demonstrates a partial attempt to form the induction circuit, even though the architecture lacks the necessary second layer to properly complete and leverage this mechanism for full accuracy gains. This recontextualizes the phenomenon, showing that the precursor dynamics **exist even in the shallow regime**.
>
> - Contrasting Mamba Dynamics: In stark contrast, the 1-layer Mamba also displays a similar loss bump, yet it successfully solves the task (99% accuracy). This outcome is a fundamental and novel distinction in architectural capability: it suggests that the SSM's expressive power allows it to compress the function required for associative recall, which necessitates a two-layer circuit in a Transformer, into its wide, single-layer recurrent state (given the presence of a convolution). This distinction in how expressive power interacts with training dynamics is a core novelty of our work.
>
> -------
>
> We believe these clarifications give a clearer understanding of the phenomenon shown in the paper. Should the Reviewer have any further questions or require additional analysis, we would be happy and open to provide them.

---

> > ### Comment · Reviewer_3xjD · 2025-11-27
> >
> > I thank the authors for their response. Ultimately, I still think that the lack of a more fundamentally sound explanation is a limiting factor that leaves the paper feeling more speculative than anything else. Given the synthetic nature of the tasks being experimented, I believe that developing a more unique perspective of the problem and validating it is one possible direction for improving this work, or giving more sound theoretical backing to the conclusions could also be a more substantial contribution than what is currently presented. Accordingly, I would prefer to maintain my current score.

---

> > > ### Author Response · Authors · 2025-11-28
> > > **Official comment by Authors**
> > >
> > > We thank the reviewer for their final feedback.
> > >
> > > We acknowledge the desire for theoretical proofs. However, we posit that empirical diagnosis is an equally crucial step in the scientific process. Before our work, the community lacked a clear understanding of why SSMs often struggle with recall, specifically, distinguishing between expressivity limits (capacity) and learnability issues (optimization).
> > > Our results conclusively identify **optimization instability as the root cause**. By isolating this phenomenon, we provide the necessary groundwork that future theoretical work will require.
> > >
> > > We hope the reviewer recognizes this empirical diagnosis as a significant contribution to the field.

---

### Official Review · Reviewer_XvGh · 2025-10-31

**Soundness:** 2
**Presentation:** 2
**Contribution:** 2
**Rating:** 2
**Confidence:** 3

**Summary:**

This paper conducts an empirical study on SSMs and Transformers in terms of their ability to solve associative recall and copying. For both tasks, the authors find that Transformers are more robust whereas SSMs are more sensitive to the choice of hyperparameters. Focused on the associative recall task (MQAR), the authors empirically show that Transformer benefits from scaling depth whereas SSMs benefit from scaling width. Consequently, on single-layer models, Transformer fails to solve MQAR whereas SSMs succeed, with Mamba being more efficient than Hyena. The author provide ablations on the architectural components in Mamba and Transformers, showing that removing convolution or gating in Mamba still retain its performance on MQAR.

**Strengths:**

1. Understanding the optimization difference between SSMs and Transformers is an important and timely research direction.

2. The authors conduct extensive experiments, with a fine-grained grid search over learning rate hyperparameters.

**Weaknesses:**

1. Although the paper presents empirical evidence on the optimization difference between SSMs and Transformers, it lacks theoretical analysis or discussions to explain such observations. Some theoretical grounding can also be found in [1] [2] (which give concrete constructions on 2-layer Transformers and 1-layer Mamba solving associative recall, respectively).

2. Some claims are not well supported, such as the optimization difference in SSM and transformers (see Question 1) and the benefit of width in SSM (see Question 2).

References:

[1] Bietti et al. "Birth of a transformer: A memory viewpoint." NeurIPS 2023.

[2] Huang et al. "Understanding Input Selectivity in Mamba: Impact on Approximation Power, Memorization, and Associative Recall Capacity." ICML 2025.

**Questions:**

1. Transformers interleave attention mixer layer and (pointwise) MLP layer, whereas Mamba is a stack of Mamba block containing a (short) convolution layer, a SSM mixer layer, and a gating branch. Can the optimization difference, specifically the learning rate sensitivity of Mamba, arise from its additional degrees of freedom (e.g., the choice of convolution kernel size, the state matrix, and gating branch)? To strengthen the claims, can the authors fix similar backbone (say using Mamba block), and study the learning rate sensitivity of a softmax attention layer versus the original Mamba S6 layer?

2. In Sec 4.1, the authors claim that recurrent models always benefit from width (line 312). However, from Fig.3 (sequence length 256, KV pairs 16), we see that Mamba with large width (512, 2048) works less well than smaller width (64-256). Moreover, when datasets contain noise, large-capacity models may overfit, resulting in the bias-variance tradeoff. Can the authors clarify or/and suitably modify their claim?

3. Sec 7. The role of convolution in 1-layer Mamba for solving MQAR. The authors find that removing the convolution in 1-layer Mamba does not affect its ability to solve MQAR. I find this result surprising, contradicting the empirical findings in [3] where they showed that convolution kernel in Mamba with length $\geq 2$ is necessary for 1-layer Mamba to solve MQAR (Fig 6). Can the authors explain possible solution mechanisms and/or provide evidence? Can the authors examine how robust is the Mamba without convolution for solving MQAR when increasing the sequence length, key-value pairs, etc?

References

[3] Arora et al. "Mechanistic evaluation of Transformers and state space models." arXiv:2505.15105 (2025).

---

> ### Author Response · Authors · 2025-11-24
> **Official comment by Authors**
>
> We thank the Reviewer for their time, for recognizing the importance and timeliness of our work. Your review and the resources you provided are greatly appreciated, as they helped us frame our contribution more precisely within the ongoing research. We acknowledge the resources provided (Bietti et al., 2023, ,Huang et al., 2025 and Arora et al. 2025) as highly valuable additions that investigate how Attention and Mamba mechanistically solve the Associative Recall task. We incorporated them into our discussion. However, we clarify that our paper's primary focus is on **learnability** rather than raw **expressivity** (what models can do), tackling the preceding practical question of "can that solution be found in practice?". Our core thesis is that optimization instability is the main differentiator.
>
> --------
>
> 1. Response to Lack of Theory / Backbone Ablation
>
> The reviewer questioned the lack of theoretical analysis and asked if Mamba's instability arises from its additional components (gating, convolution).
> - Our Core Thesis: Our paper provides the empirical diagnosis that optimization instability (brittleness, Fig. 1) is the root problem. We rely on prior literature that established the expressive potential of these models, but focus on the "learnability step".
> - Ablation Confirmation with Existing Experiments: The reviewer suggested studying the Mamba backbone. We confirm that **the extensive ablations of Mamba's core structure you suggested were performed and are already in the paper in Table 2**:
> The experiments with Mamba w/o gating and Mamba as a Transformer (S6+MLP) confirm that Mamba's initial performance and **optimization brittleness** (narrow LR window) **are preserved**. This suggests the optimization issue is not an artifact of auxiliary components but is inherent to the core S6 mixing mechanism.
>
> 2. Correction: The Role of Convolution
>
> The reviewer questioned our previous finding that convolution was redundant, citing Arora et al. (2025). This was a point that led to a correction:
> - Corrected Finding: We re-validated data and show that the convolution is necessary for the 1-layer Mamba to succeed. Mamba without conv1d fails the task (0-2% accuracy) and does not improve with increased width, becoming **mechanistically equivalent to the 1-layer Transformer** as shown in the updated Table 2.
> - The Unified Insight: This correction strengthens our framework: The convolution is an architectural component that grants the SSM the necessary locality/expressivity for recall in the 1-layer setting, but **still maintains its optimization brittleness**, which remains our principal finding.
>
> -------
>
> 3. Width vs. Depth Scaling Claim
>
> The reviewer questioned the benefit of width, pointing to performance dips (Fig. 3) and referencing literature that prioritizes the hidden state (Orvieto et al. 2024, Jelassi et al. 2024).
> - Clarification on Scaling: We agree with the references “SSMs rely on a larger hidden state”, but our point is about **prioritization**. Our copy task (Table 1) proves that matching parameter counts requires prioritizing width over depth (12-layer wide Mamba succeeds, 24-layer narrow Mamba fails).
> - Recontextualizing Dips: The observed dips in performance at high width (Fig. 3, Seq 256) are direct evidence of our central thesis. Since in our experiments we used the same compute budget, this is an optimization failure, where the narrow LR window for those complex configurations was slightly missed, strengthening our main claim.
>
> -------
>
> 4. Conclusion on Implementation
>
> We agree that understanding the precise role of each architectural component is crucial for discerning what drives capability versus what drives stability. Methodologically, we emphasize that many architectural choices were intentionally kept consistent with the Zoology codebase (Arora et al., 2023) to ensure comparability and fair analysis across models, only deviating when necessary for specific ablations (e.g., studying the role of convolution, depth/width scaling). Crucially, the persistence of the optimization challenge is universally confirmed: our ablation experiments show the narrow LR window remains even with structural changes to Mamba, and our new results with 12-layer stacks confirm that Mamba's inherent instability persists in deep regimes. While the Transformer's stability is also slightly reduced in this massive scale, this merely confirms that optimization control is a major challenge for all deep networks, further emphasizing that Mamba's persistent brittleness is a fundamental, architectural property that defines its learnability.
>
> ---------
>
> We believe these clarifications, anchored by the crucial new insight on the convolution, resolve the issues of theoretical ambiguity. Should the Reviewer have any further questions or require additional analysis, we would be happy and open to provide them.

---

> > ### Comment · Reviewer_XvGh · 2025-11-26
> >
> > I thank the authors for their detailed responses. Quick follow-up:
> >
> > > Response to Lack of Theory / Backbone Ablation: ...the extensive ablations of Mamba's core structure you suggested were performed and are already in the paper in Table 2:...
> >
> > * To clarify, does "attention" mean 1-layer softmax attention layer followed by 1-MLP?
> > * Even if this is the case (so that we can compare "attention" and "S6+MLP" directly), table 2 only provides the final performance, not showing the optimization brittleness from narrowing window of learning rate? Do I miss anything?

---

> > > ### Author Response · Authors · 2025-11-28
> > > **Official comment by Authors**
> > >
> > > We thank the reviewer for the follow-up and the opportunity to clarify.
> > >
> > > - Definition of Attention: You are correct. In Table 2, "Attention" refers to the standard Transformer block structure: a sequence mixer layer (Softmax Attention) followed by a standard MLP. This definition of a "layer" (Mixer + MLP) applies to all comparable models in our study, with the exception of the standard Mamba block which integrates these components differently.
> > >
> > > - Brittleness in Ablations: We confirm that the **optimization brittleness persists** in all the ablated Mamba models, including the "Mamba as Transformer" (S6+MLP) variant. In Table 2, we deliberately reported only the peak accuracy to isolate the question of expressivity (proving that the architecture can solve the task) since we were trying to make the two architectures closer to each other. The choice is deliberate since the plot learning rate against accuracy would have been with a lot of overlaps and visually less significant than a Table. However, our result confirms that the learnability remains constrained to the characteristic narrow learning rate window of SSMs.
> > >
> > > We hope this clarifies our claims and definitions.

---

### Official Review · Reviewer_1bHM · 2025-10-31

**Soundness:** 3
**Presentation:** 4
**Contribution:** 3
**Rating:** 4
**Confidence:** 4

**Summary:**

The paper empirically compares state-space models (SSMs) and Transformers on associative recall (MQAR) and copying. It finds SSMs succeed only within a narrow learning-rate window, whereas Transformers are robust across a wide range; this optimization brittleness can confound expressivity comparisons. It further shows opposite scaling preferences—SSMs benefit from width, Transformers from depth—and reports *1-layer* training dynamics (e.g., loss “bumps” reminiscent of induction circuits) that differ across families.

**Strengths:**

### Originality

* Re-frames SSM vs. Transformer comparisons through learnability: finds critical LR-sensitivity for modern SSMs on MQAR/copying, contrasting with robust Transformers.

### Quality

* Executes large LR sweeps and reports results (3 seeds) that expose narrow “goldilocks” regions for Mamba/Hyena, and wide basins for attention.
* Provides scaling experiments showing width helps SSMs while depth helps Transformers; includes ablations (conv on Q/K/V; gating) that isolate drivers.

### Clarity

* Figures and captions explicitly note seed counts and LR-sweep setup.

### Significance

* Highlights **optimization stability** as the key differentiator; argues scaling along each architecture’s “preferred axis” (width vs. depth) is crucial for fair comparisons.

**Weaknesses:**

1. **Most results center on 1–2-layer models on synthetic tasks**. It’s unclear if the LR brittleness persists for deeper stacks (e.g., 12–24 layers) or on standard LM tasks. Adding at least one real-world benchmark (e.g., small LM perplexity) would be better.

2. Many plots average over 3 seeds; for **stability** claims, 3 is thin.

3. Modern recurrent models encompass many SSM (and non-SSM) variants; so the paper should name explicit implementations used in each figure (e.g., Mamba, Hyena, RWKV, H3) in-line, and clarify the Based attention baseline.

**Questions:**

1. Can you replicate Fig. 3 on a 12-layer Mamba-style stack and report the LR-window width vs. depth? This would test whether brittleness is primarily a shallow-model artifact.

2. For Fig. 1–4/7, please re-run with ≥5 seeds and plot CIs; this will calibrate how “narrow” the SSM LR window is statistically.

---

> ### Author Response · Authors · 2025-11-24
> **Official comment by Authors**
>
> We thank the Reviewer for their highly insightful and constructive review. We are particularly grateful that the reviewer recognized the originality and significance of our work on optimization stability, which we agree is the key differentiator between these architectures. We have addressed the reviewer's main concerns regarding model depth, seed count, and implementation clarity, and the results have led to important clarifications that strengthen our conclusions.
>
> --------
>
> 1. On Using 1-2 Layers vs. Deeper Stacks
>
> We agree that the persistence of brittleness in deeper stacks is an interesting question. We maintain that focusing on 1-2 layers is essential for **mechanistic isolation**, as using deep models on synthetic tasks introduces overwhelming confounds and obscures the minimal necessary circuit. To address the core of the reviewer’s query:
> - New 12-Layer Experiments: We have run new experiments on 12-layers Mamba and Transformer stack (as requested) and confirm that the learning rate window for Mamba **remains critically narrow**, as shown in Figure 13 (Appendix A6). Regarding training perplexity in transformers vs SSMs, it is known in the literature that Mamba does not have shortcomings compared to Attention. This is however, not a good indicator for capabilities: as well-known, e.g. from MMLU experiments in Hendrycks et al., 2021, Mamba lacks fundamental capabilities associated with in-context recall even when reaching the same pretraining perplexity of a transformer.
> - Evidence from Copy Task: Our data on the Copy Task (Table 1) already uses 12- and 24-layer models and strongly supports our claim that SSMs scaling must prioritize width over depth.
>
> -------
>
> 2. On Statistical Rigor and Clarity
>
> We appreciate the reviewer’s emphasis on empirical rigor and agree that strengthening our statistical base is valuable.
> - Seed Count: To fully satisfy this request, we have re-run the key experiments with 5 seeds and confirm that the results and **conclusions are unchanged**; the narrow window of success for SSMs remains robust. This evidence confirms our initial methodological decision: Our central claim is one of instability. While proving a model's stability requires numerous seeds to minimize variance, proving instability is sufficiently demonstrated by consistent failure, as observed in our initial runs. The fact that the performance drops deterministically outside a narrow band underscores the brittle nature of the optimization landscape, which is not a statistical artifact.
> - Clarity of Implementation: We agree that clear implementation details are essential for reproducibility. We added in the appendix all the implementations and explicitly named the model variants, ensuring full transparency. Given the restrictions on the page limits, it was not possible to add those in-line, but we will do it in the final version of the paper. Crucially, we will highlight that all these implementations are built upon the shared, open-source codebase provided by the Zoology work (Arora et al., 2023), maintaining full consistency with that established benchmark.
>
> -----
>
> We are confident that these new changes and insights, derived directly from the questions raised by the reviewer, result in a substantially more robust and rigorous paper. Should the Reviewer have any further questions or require additional analysis, we would be happy and open to provide them.

---

> > ### Author Response · Authors · 2025-11-28
> > **Official comment by Authors**
> >
> > Dear Reviewer,
> >
> > As the discussion period draws to a close, we wanted to confirm that our new results have addressed your concerns regarding depth and statistical robustness.
> > We added 12-layer experiments (Figure 13 in Appendix A.6 ) and re-ran our key analyses with 5 seeds. These results confirm that the optimization brittleness is not a shallow artifact but persists in the deep regime, which we believe significantly strengthens the paper's core claims.
> >
> > We hope this response clarifies your doubts and we are happy to provide any further clarifications if needed.

---

### Official Review · Reviewer_p7X8 · 2025-11-01

**Soundness:** 3
**Presentation:** 3
**Contribution:** 3
**Rating:** 6
**Confidence:** 4

**Summary:**

This paper analyzes the differences between the transformer model architecture and the state space model (SSM), one type of recurrent model architecture, in associated recall and copy tasks. The paper finds that compared to the transformer model, the SSM suffers from severe optimization instability, and training with different learning rates is crucial for the performance evaluation of the SSM. The paper also reveals the distinct scaling behaviors and induction head formation mechanisms of the two architectures.

**Strengths:**

1. This paper demonstrates the differences between SSM and transformer architectures from the perspective of model optimization and stability, providing a new dimension for understanding and analyzing SSM models.
2. The experimental analysis is very interesting, especially in exploring in-context recall-intensive tasks that are a key focus of these linear architectures, which may bring important insights to the development of the RNN community.
3. The paper is well-written, clearly structured, and easy to understand.

**Weaknesses:**

1. The paper doesn't adequately explain why even a single-layer Mamba without conv1d can still perform well on MQAR. It would be better to analyze the formation mechanism of induction heads in single-layer recurrent models.
2. The paper presents the various differences between the transformer and SSM in a fragmented manner, seemingly lacking a unified analytical explanation. Is there a connection between the optimization instability, width/depth scaling behavior, and induction heads phenomenon mentioned in the paper?

**Questions:**

See weaknesses

---

> ### Author Response · Authors · 2025-11-24
> **Official comment by Authors**
>
> We thank the Reviewer for their thoughtful evaluation and for highlighting the optimization and stability perspectives as core strengths of our work. We are pleased that the reviewer believes it can be interesting and insightful for the RNN and SSM community. Below, we address the reviewer's concerns and valuable questions.
>
> ------
>
> 1. The Role of Architectural Components: Correction and Clarification
>
> The reviewer correctly asked for a better explanation of the role of the conv1d component. Our subsequent analysis led to a small correction that actually strengthens our position. Mamba without convolution becomes **mechanistically closer** to a Transformer.
> Indeed, a 1-layer Mamba without conv1d fails the task (as shown in the corrected Table 2), and its performance does not improve even with increased width in the single-layer setting. The capability only recovers at 2 layers. This behaviour perfectly mirrors the observed behavior of the Transformer.
> This however strengthens our position, because also in this setting, Mamba **still preserves the optimization brittleness** of SSMs, which remains the main point of our analysis.
>
> -----
>
> 2. Unifying the Analytical Explanation
>
> We agree that the core connection between our experiments must be made explicit. Our entire empirical investigation is connected by the diagnostic task of MQAR, which initially revealed the optimization brittleness of recurrent models. This discovery then compelled us to investigate the resulting architectural differences in scaling and mechanisms, as outlined below:
> - (A) Core Discovery: Optimization Brittleness as the Initial Insight
>
> The study of MQAR first revealed that modern recurrent models (SSMs) suffer from a critical optimization instability, with success confined to a narrow window of learning rates (Fig. 1). This finding is significant because:
> - - It proved that SSMs behave fundamentally differently during training than Transformers and are actually closer to the training of RNNs.
> - - It identified a persistent confounding variable that must be controlled for in any fair model comparison.
>
> This initial insight demanded a comprehensive investigation into how SSMs and Transformers must be **architecturally configured** to successfully solve memory tasks, leading to the subsequent analysis of scaling and expressive mechanisms.
>
> - (B) The Consequence: Differential Scaling for Architectural Success
>
> Since the models exhibit differential learning stability, their intrinsic architectural limitations dictate their path to success on AR, requiring different scaling strategies:
> - - SSM's Necessity: Width: To solve memory tasks, SSMs inherently rely on a finite hidden state. This constraint compels them to leverage **width** to ease the task of information compression.
> - - Transformer's Necessity: Depth: The Transformer, being robust to optimization, instead relies on **depth** to build complex multi-layer circuits for recall.
>
> which leads to the final specific investigation of the shallow transformer.
>
> - (C) Mechanistic Validation: The Role of Depth (Induction Heads)
>
> Finally, observing the training dynamics validates the necessity of the Transformer's depth:
> - - The Transformer's preferred memory mechanism, the Induction Head circuit, is considered a **multi-layer phenomenon** (as in Olsson et al. (2022), Bietti et al. (2023)).
> - - Our novel experiment showing the characteristic loss bump in the 1-layer Transformer (Fig. 6) reveals precursor dynamics, indicating that while the signal is generated in one layer, depth is the necessary architectural component required to **fully realize** and utilize the recall mechanism for correct prediction.
>
> The optimization brittleness, therefore, is the key empirical phenomenon that connects our findings and establishes the need to treat SSMs and Transformers based on their differential scaling requirements and stability properties.
>
> ------
>
> We believe these clarifications, anchored by the crucial new insight on the convolution, give a more defined structure to our work. Should the Reviewer have any further questions or require additional analysis, we would be happy and open to provide them.

---

> > ### Comment · Reviewer_p7X8 · 2025-11-27
> >
> > The original key insight of this paper was to reveal the differences between SSM and Transformer in the training dynamics and the formation of induction heads. However, during the rebuttal process, the paper was updated with conclusions that are completely opposite to the original ones, and still lacks explanatory support. This makes me concerned about the soundness of this paper, and I have to lower my rating from 6 to 4.

---

> > > ### Author Response · Authors · 2025-11-28
> > > **Official comment by Authors**
> > >
> > > We thank the reviewer for their continued engagement. We understand the concern regarding the update on the convolution ablation. However, we wish to clarify exactly what changed and, crucially, what remains unchanged in light of this new result.
> > >
> > > - What Changed (Expressivity Mechanism): Originally, we observed 1-layer Mamba performing well. Upon re-verification for the rebuttal, we identified that the Conv1d layer is the specific driver of this success. This corrects the mechanistic understanding of **how** the task is solved: Mamba relies on the local processing of the convolution to act as a pseudo-induction head in a single layer.
> > >
> > > - What Remains Unchanged (Optimization Instability): Our **paper’s central finding** is not about how Mamba solves the task (expressivity), but the fact that it is **hard to train** (optimization). This finding **holds true across all settings**: Mamba without Conv1d, Mamba as a Transformer (S6+MLP), Mamba without gating, and our new 12-layer experiments all share the same narrow learning rate window.
> > >
> > > The correction regarding the convolution isolates the source of expressivity, but it confirms that the **instability is inherent to the recurrent mixing mechanism** (S6), regardless of the surrounding architectural components.
> > >
> > > We believe the paper is now scientifically more accurate, and the diagnosis of optimization instability remains a robust and novel contribution. We hope this clarification resolves your concerns and respectfully ask the reviewer to reconsider the score given this clarification.

---

### Author Response · Authors · 2025-11-28
**Official final remark by Authors**

We thank all reviewers for their engagement, which has significantly improved the rigor of our paper. Through the discussion phase, we have:

- Confirmed Instability in Deeper Models. We conducted new experiments on 12-layer models (Appendix A.6), confirming that the optimization brittleness  (the narrow learning rate window) **is not an artifact of shallow networks** but a persistent characteristic of SSMs.

- Disentangled Expressivity from Learnability. We re-ablated our models to clarify that while the Conv1d layer is essential for the expressivity of a 1-layer Mamba , our data confirms that the optimization instability remains regardless of this component. This reinforces our core finding: **the brittleness stems fundamentally from the recurrent nature of the sequence mixer** (S6), aligning its training dynamics more closely with RNNs than Transformers.

- Unified the Narrative. Our work provides a clear empirical diagnosis: SSMs and Transformers differ fundamentally in learnability. Transformers are robust and scale effectively with depth, whereas SSMs are brittle and require scaling in width to succeed.

While we acknowledge the desire for theoretical proofs, we posit that this paper serves as an **empirical warning**. We demonstrate that optimization stability can be a relevant confounding factor in SSM evaluation. Without the precise tuning we demonstrate, future evaluations of efficient architectures risk being fundamentally misaligned.

---

### Author Response · Authors · 2025-12-01
**Final comment for Area Chair**

To the Area Chair,

We sincerely thank the Area Chair for the additional effort and time dedicated to overseeing our submission, especially given the current situation that affected the rebuttal period. To assist in your final assessment and minimize your workload, we have synthesized the entire discussion history below. We summarize our paper's core contributions, the consensus regarding strengths and weaknesses of our manuscript, and how we have systematically addressed every concern raised during the rebuttal. For space constraint, we divided the response into two parts.

----

**1. Summary of Contributions**

Our work provides a critical **empirical** diagnosis of the practical differences between modern recurrent models and Transformers in solving synthetic reasoning tasks, in particular Multi-query associative recall (MQAR) and copying:

- **Learnability Diagnosis:** We reveal that modern SSMs (e.g., Mamba, Hyena) suffer from critical **optimization instability**, where success is confined to a dangerously narrow learning rate window. This shift the discussion on the difference and similarities between SSMs and Transformers: from an **expressivity problem** (if the architecture can solve the task) to a **learnability problem** (how hard is it to learn the task)

- **Scaling Dichotomy:** We demonstrate that to solve tasks like MQAR and Copying, SSMs must prioritize scaling in **width** (to ease compression into the hidden state), whereas Transformers rely on **depth** to build the necessary circuit to recall information in-context (induction heads). This fact, in addition to the optimization instability, brings the family of SSMs architecture closer to the general family of RNNs.

- **1-Layer Dynamics:** We uncover that while 1-layer Transformers cannot solve the task, they exhibit **loss dynamics resembling induction head formation**, a precursor mechanism usually associated with deeper models [1], [2].

- **Empirical Warning:** We argue that this optimization brittleness is a major **confounding factor**, and future evaluations of efficient architectures risk being invalid if they do not account for this stability gap [3], [4].

---

**2. Strengths (Consensus across Reviewers)**

Reviewers acknowledged the timeliness and significance of the empirical findings for the SSMs and recurrent models community:

- **Timeliness & Significance:** Reviewers agreed that understanding the optimization differences between SSMs and Transformers is an important and timely research direction.

- **Originality of Learnability Focus:** Reviewers appreciated re-framing the comparison through the lens of **learnability and optimization stability** rather than just expressivity.

- **Extensive Empirical Scope:** The analysis was praised for its extensive grid searches (~20k GPU hours) that exposed the narrow regions of suitable learning rates for Mamba/Hyena versus the wide basins for Attention.

- **Valuable Scaling Insights:** Reviewers highlighted the actionable insights regarding the opposing scaling preferences (width vs. depth) for different architectures.

---

> ### Author Response · Authors · 2025-12-01
> **Final comment Area chair (pt. 2)**
>
> **3. Weaknesses (Synthesized) & Rebuttal Actions**
>
> We addressed every major weakness raised by reviewers (1bHM, XvGh, p7X8, 3xjD) through new experiments and clarifications.
>
> **A. Methodology & Generalization (Depth & Seeds)**
>
> - **Critique:**
> Reviewer 1bHM questioned whether our findings were artifacts of shallow (1-2 layer) networks and requested more seeds for statistical rigor.
>
> - **Action:**
>
> - - **Deep Models:** We conducted new experiments on 12-layer models (Appendix A.6), confirming that the critical **optimization brittleness persists even in deep regimes**. We, however, highlight that the choice of using shallower models was deliberate, to match the setting of previous works [1], [3], and to isolate as much as possible the effect of every component of the architecures.
>
> - - **Statistical Rigor:** We re-ran key experiments with 5 seeds, confirming that the narrow learning rate window is deterministic and statistically robust in SSMs.
>
> **B. Mechanistic Clarity (Role of Convolution)**
>
> - **Critique:**
> Reviewer XvGh noted that 1-layer Mamba requires the Conv1d layer for expressivity [5], [6], asking for clarification on its role.
>
> - **Clarification:** We performed ablations (Table 2) confirming that removing Conv1d causes 1-layer Mamba to fail, effectively behaving like a 1-layer Transformer. Crucially, we showed that the **optimization instability persists** regardless of the backbone configuration (S6, S6+MLP, or Mamba block), reinforcing our core thesis.
>
> **C. Contradiction of Conv1d (Reviewer p7X8)**
>
> - **Critique**: Following our correction regarding the Conv1d layer, reviewer p7X8 perceived it as a contradiction to our key insight.
>
> - **Our Response**: This appears to be a misunderstanding. Our paper is a diagnosis of optimization stability, not just expressivity mechanism. The correction regarding convolution clarified how the task is solved, but confirmed our central finding: **Mamba remains critically harder to train** than Transformers across all settings, strengthening our core finding.
>
> **D. Theoretical Backing (Reviewers XvGh & 3xjD)**
>
> - **Critique:** Some reviewers requested theoretical proofs for why this instability occurs.
> - **Action:** We argued that scientific progress requires **accurate empirical diagnosis** first. By rigorously isolating optimization brittleness from capacity limits,we provide the necessary empirical groundwork that future theoretical analysis must build upon.
>
> ---
>
> We believe the paper now offers a robust, deeply validated warning for the community: efficient modeling research must treat optimization stability as a first-class objective.
>
> ---
>
> References
>
> [1] Olsson et al. "In-context learning and induction heads" (2022)
>
> [2] Bietti et al. "Birth of a transformer: A memory viewpoint" (2023)
>
> [3] Arora et al. "Zoology: Measuring and improving recall in efficient language models" (2023)
>
> [4] Jelassi et al. "Repeat after me: Transformers are better than state space models at copying" (2024)
>
> [5] Ali et al. "The hidden attention of Mamba models." (2024).
>
> [6] Arora et al. "Mechanistic evaluation of Transformers and state space models." (2025).

---

### Meta-Review · Area_Chair_XsL7 · 2025-12-29

**Summary:**

This submission presents an extensive empirical study comparing state space models (SSMs) and Transformers on associative recall and copying tasks, with a central claim that modern SSMs suffer from severe optimization brittleness, manifested as a narrow learning rate window, whereas Transformers are substantially more robust. The topic is timely, and several reviewers agree that framing the comparison through learnability and optimization stability is interesting and potentially important. The experimental effort is non-trivial, with careful learning rate sweeps, ablations, and additional experiments added during the discussion phase, including deeper (12-layer) models.

Reviewers agreed that the paper reports an interesting empirical observation on optimization brittleness in SSMs. However, several consistent concerns led to the final decision.

The primary issue is the lack of a clear and well-supported explanatory or theoretical foundation. While framed as an empirical diagnosis, the paper does not convincingly connect optimization instability, width versus depth scaling, and proposed mechanistic interpretations, leaving key claims largely descriptive and speculative. The narrative was also seen as fragmented.
A second concern is the limited empirical scope. Experiments focus mainly on MQAR and closely related synthetic tasks, with insufficient validation on realistic or standard language modeling benchmarks, making the generality and practical impact unclear.
Finally, revisions during discussion, including corrections on the role of convolution in single-layer Mamba, improved accuracy but raised concerns about soundness and shifting interpretations.
Overall, despite its merits, the paper lacks sufficient theoretical grounding, a unified explanation, and breadth of evaluation to meet the acceptance bar.

**Reviewer Concerns:**

I do find that the paper proposes an interesting finding based on the assumption and experiments. Despite the merits, the paper does not meet the bar for acceptance.

Across reviewers, a consistent concern is the lack of a solid explanatory or theoretical foundation. While the authors emphasize that the work is intended as an empirical diagnosis rather than a theoretical contribution, the absence of a unifying, well-supported explanation significantly weakens the paper. Many claims remain descriptive and speculative, especially given the synthetic nature of the tasks studied, and several reviewers found that the narrative feels fragmented even after the rebuttal. The connections between optimization instability, width versus depth scaling, and mechanistic phenomena such as induction heads are asserted rather than convincingly established.

A second major issue is scope and benchmarking. The empirical evidence is heavily concentrated on MQAR and closely related synthetic tasks, with limited validation on more realistic or standard language modeling benchmarks. While additional depth experiments and more seeds were added, these remain confined to the same narrow task family. As a result, it is unclear how general or impactful the conclusions are for practical model evaluation or deployment. Multiple reviewers explicitly noted that broader benchmarks or stronger evidence beyond synthetic recall tasks are necessary to support the paper’s broader claims.

Finally, the discussion process revealed instability in the paper’s conclusions themselves. Corrections regarding the role of convolution in single-layer Mamba, while appreciated for improving factual accuracy, led at least one reviewer to lower their score due to concerns about soundness and shifting interpretations. Although the authors argue that the core optimization claim remains unchanged, this evolution further underscores the lack of a firm conceptual grounding.

In summary, while the paper raises an interesting and potentially important observation about optimization brittleness in SSMs, the current version lacks sufficient theoretical justification, unified explanation, and breadth of evaluation to support its strong claims. Given these limitations and the overall reviewer consensus trending below the acceptance threshold, the final decision is **reject**.

**Reviewer Scores:**

Reviewer p7X8: This reviewer explicitly revised their score downward during the discussion, from 6 to 4, citing concerns that the updated conclusions contradicted the original framing and still lacked sufficient explanatory support. With full participation, the final score would remain at 4.

Reviewer 1bHM: This reviewer raised concrete concerns about depth, seed count, and benchmark breadth, many of which were partially addressed by additional experiments. However, the reviewer already indicated ambivalence and rated the paper at 4. With full participation, the score would likely remain unchanged at 4, as the core concern about limited benchmarks and scope is not fully resolved.

Reviewer XvGh: This reviewer consistently emphasized the lack of theoretical grounding and questioned whether the empirical evidence sufficiently supports the optimization claims. Despite detailed rebuttals, their stance remained skeptical. With full participation, the score would remain at 2.

Reviewer 3xjD: This reviewer maintained a strong position that the work is overly speculative and insufficiently grounded, even after the authors’ clarifications. With full participation, the score would remain at 2.

Overall, while some reviewers acknowledged the added experiments and clarifications, none would have increased their scores meaningfully after full discussion, and one reviewer explicitly decreased their score.

---

### Decision · Program_Chairs · 2026-01-26

Reject